

# Time-series analysis of meteorological factors and emergency department visits due to dog/cat bites in Jinshan area, China

Pei Pang[1], Xiaoyong Zhou[1,2], Yabin Hu[3,4], Yin Zhang[5], Baoshi He[6] and Guoxiong Xu[7]

[1] Department of Medical Affairs, Jinshan Hospital, Fudan University, Shanghai, China
[2] Emergency Department, Jinshan Hospital, Fudan University, Shanghai, China
[3] Key Lab of Health Technology Assessment, National Health Commission of the People's Republic of China, Fudan University, Shanghai, China
[4] Key Laboratory of Public Health Safety, Ministry of Education, School of Public Health, Fudan University, Shanghai, China
[5] Shanghai Meteorological Service Center, Shanghai, China
[6] Key Laboratory of Watershed Geographic Sciences, Nanjing Institute of Geography and Limnology, Chinese Academy of Sciences, Nanjing, China
[7] Research Center for Clinical Medicine, Jinshan Hospital, Fudan University, Shanghai, China

Corresponding authors
Xiaoyong Zhou,
zhou_xy@fudan.edu.cn
Guoxiong Xu,
guoxiong.xu@fudan.edu.cn

## ABSTRACT

**Background**. Meteorological factors play an important role in human health. Clarifying the occurrence of dog and cat bites (DCBs) under different meteorological conditions can provide key insights into the prevention of DCBs. Therefore, the objective of the study was to explore the relationship between meteorological factors and DCBs and to provide caution to avoid the incidents that may occur by DCBs.

**Methods**. In this study, data on meteorological factors and cases of DCBs were retrospectively collected at the Shanghai Climate Center and Jinshan Hospital of Fudan University, respectively, in 2016–2020. The distributed lag non-linear and time series model (DLNM) were used to examine the effect of meteorological elements on daily hospital visits due to DCBs.

**Results**. A total of 26,857 DCBs were collected ranging from 1 to 39 cases per day. The relationship between ambient temperature and DCBs was J-shaped. DCBs were positively correlated with daily mean temperature (rs = 0.588, $P < 0.01$). The relative risk (RR) of DCBs was associated with high temperature (RR = 1.450; 95% CI [1.220–1.722]). Female was more susceptible to high temperature than male. High temperature increased the risk of DCBs.

**Conclusions**. The extremely high temperature increased the risk of injuries caused by DCBs, particularly for females. These data may help to develop public health strategies for potentially avoiding the occurrence of DCBs.

# INTRODUCTION

Dog and cat bites (DCBs) are now serious social and public health problems globally because the incidence of DCBs continues to rise (*Campagna et al., 2023*; *Loder, 2019*; *Roman et al., 2023*). In the United States, DCBs account for approximately 1% of all emergency department (ED) visits and an estimated 4.5 million dog bites and 0.4 million cat bites occur every year (*Bula-Rudas & Olcott, 2018*; *Maniscalco & Edens, 2023*). In 2018, the WHO survey data showed that 76–94% and 2–50% of animal injuries are caused by dogs and cats (*Williams, 2018*). In China, an estimated 40 million people are injured by dogs and cats every year (*Liu et al., 2017*). In six provinces (Anhui, Guangxi, Guizhou, Hunan, Jiangsu, and Shandong) of China, 1,018,367 people were treated in clinics due to dog injuries in 2016 (*Li et al., 2018*).

The outcomes of DCBs on human health include varying degrees of physical and psychological consequences, including bites, scratches, secondary infections, surgeries and sequelae, and post-traumatic stress disorder (PTSD) (*Cianciara, Gorynski & Seroka, 2022*; *Giovannini et al., 2023*; *Murphy & Qaisi, 2021*). There is a high risk of contracting rabies after being injured by virus-carrying dogs. A total of 99% of human rabies infections are transmitted by dogs (*Fooks et al., 2017*; *Liu et al., 2021*). Rabies virus infections kill tens of thousands of people every year and more than 95% of human deaths occur in Asia and Africa (*Abela-Ridder et al., 2016*; *Jane Ling et al., 2023*), which impose costs on the health system and those affected consequences (*Barrios et al., 2021*).

DCBs can occur in private or public places and urban or rural areas. The risk factors are various, including the dog or cat, the owner or victim, the natural environment, and other factors (*Bay et al., 2021*; *Campagna et al., 2023*; *Chen et al., 2018*). Indeed, many issues can be considered for minimizing the risk factors such as specific factors (sex, castration/spay status, breed), victim factors (age, gender, familiarity with dog and cat, victim behavior), the relationship between the victim and the animals, the time and place of occurrence, meteorological factors, air pollution factors, *etc* (*Caffrey et al., 2019*; *Chevalier et al., 2021*; *Park et al., 2019*; *Zangari et al., 2021*)

Most earlier studies focused on the social and cultural aspects that contribute to DCBs and are associated with misbehaviors by humans or animals, *e.g.*, invading their territory, breeding, interfering with their eating, neglecting to vaccinate them, *etc* (*Matthias et al., 2015*; *Patronek et al., 2013*). Currently, a growing number of studies have focused on links between meteorological factors and various injuries, but few on DCBs (*Chen et al., 2019*; *Lippmann et al., 2013*; *Oh et al., 2020*). The continued rise in global ambient temperatures has emerged and the impact of meteorological factors on health outcomes varies by geographic region and population (*Baccini et al., 2011*; *Ma et al., 2015*). This study aims to provide public health strategies that may avoid the occurrence of DCBs and ultimately prevent the resulting incidents.

## MATERIALS & METHODS

### Study area

The research was conducted in the Jinshan area located in the southwest of Shanghai, which belongs to a northern subtropical region with a monsoonal climate (*Government of Jishan District, 2022*).

### Data collection

Daily records of DCBs were collected during the years 2016–2020 from Jinshan Hospital of Fudan University, the largest hospital in the Jinshan District appointed for dog bites and rabies immunization by the local government. Gender, age, principal statement, doctor's diagnosis, and registration time of cases were collected from the electronic medical records (EMR) of the hospital. Data on meteorological factors for the same period, including ambient temperature, atmospheric pressure, mean relative humidity, wind speed, precipitation, and sunshine exposure, were collected from the Shanghai Meteorological Service Center. Daily mean ambient temperature has proven to be a reliable temperature indicator and was used as the functional exposure in this study (*Sun et al., 2014*; *Wang et al., 2013*).

### Data analysis

Since the number of daily DCBs obeyed a Poisson distribution and the relationship between meteorological factors (*e.g.*, temperature) and bite events is usually nonlinear, a distributed lag non-linear model (DLNM) was used in this study to analyze the effects of meteorological factors on DCBs (*Gasparrini, Armstrong & Kenward, 2010*; *Gasparrini et al., 2015*; *Guo, Barnett & Tong, 2013*). The DLNM was utilized to control for disturbances such as humidity, insolation, long-term trends, and weekly effects. The basic pattern of the DLNM is as follows:

$$Log[E(Y_t)] = \alpha + cb(Tmean_t, maxlag = 7) + cb(DTR_t, maxlag = 7) + cb(RH_t, maxlag = 7)$$
$$+ factor(year_t) + factor(month_t) + factor(DOW_t) + factor(Holiday_t). \quad (1)$$

In the formula, t refers to the day of the observation; $E(Y_t)$ denotes estimated daily DCBs counted on day t; $\alpha$ is the intercept; cb is the "cross-basis" function for generating bi-dimensional exposure-lag response relationship with 3 degrees of freedom (df) for the exposure and lag spaces, respectively; $Tmean_t$ is the mean temperature on day $t$; "maxlag = 7" refers a maximum lag of 7 days being used to present the lagged effect of temperature; $DTR_t$ stands for diurnal temperature range on day $t$; $RH_t$ is the relative humidity on day t; factor ($year_t$) and factor ($month_t$) are used to control the seasonality and long-term trend on day t; factor ($DOW_t$) is a categorical variable for adjusting day of the week on day t; factor ($Holiday_t$) is a binary variable for adjusting public holidays in China.

We plotted exposure-response curves based on DLNM between the daily number of DCBs in hospital ED and meteorological factors. A threshold was selected and a linear-threshold model was used to quantify the effect of meteorological factors.

Because meteorological factor affects not only the ED visits for DCBs on that day but also the ED visits for DCBs in subsequent days (lag effect), we conducted a moving average lag

model to evaluate the lag effects. Here, the 0-day lag represents the meteorological factors for the current day, and the 1-day lag represents the moving average of the meteorological factors for the current day and the previous day.

After assessing the effect of environmental factors on DCBs population-wide, we repeated the same procedure to examine correlations stratified by sex and age ($\leq$14, 15–21, 22–45, 46–59, and $\geq$60 years).

Ethics approval was approved by the Ethics Committee of Jinshan Hospital (No. JIEC 2021-S38). All statistical analyses were conducted using the R statistical environment (version 3.6.3) in which the "dlnm" package was mainly used. A two-sided $P$-value of less than 0.05 was considered statistically significant.

## RESULTS

Using the house-designed technical strategy, we performed data collection and analyses of the association between DCBs and meteorological factors (Fig. S1). Table 1 shows the characteristics of 26,857 DCB cases and meteorological factors, which included 51.28% female. The number of DCBs in different age groups is listed in Table S1. Daily ED visits for DCBs were from 1 to 39 with an average of 14.7 ± 5.8. The daily mean temperature was 17.4 °C (−5.2 °C to 33.9 °C). Population aged 22–45 and 46–59 years accounted for 38.37% and 20.97%, respectively. In comparison, populations aged 0–14, 15–21, and $\geq$60 years accounted for 16.45%, 8.12%, and 16.09%, respectively. The fluctuation of daily ED visits for DCBs was consistent with the daily temperature but not with the daily atmospheric pressure (Fig. 1).

The temperature greater than or equal to 35 °C was considered hot weather. The daily maximum temperatures were mainly in July and August (Fig. S2). The month with the highest frequency of hot weather was July, followed by August and June. We found that DCBs were more likely to occur on hot days (Table S2).

Daily ED visits for DCBs were significantly correlated with daily meteorological elements except for maximum wind speed and extreme wind speed (Table 2). Meteorological elements were associated with each other significantly except mean temperature with precipitation, maximum temperature with relative humidity, and diurnal atmospheric pressure range with relative humidity. Spearman's rank correlation coefficient analyses showed that DCBs were positively correlated with daily mean temperature (rs = 0.588, $P$ < 0.01) and negatively correlated with daily mean atmospheric pressure (rs = −0.494, $P$ < 0.01).

The daily mean temperature with DCBs was J-shaped. The relative risk of DCBs was elevated with a rising temperature when the daily mean temperature was greater than 7.6 °C (17th percentile of temperature) (Fig. 2A). Females were at greater risk of DCBs over a wide range of temperatures (Figs. 2B–2C). Population aged 22–45 years were sensitive to low temperature as well as to high temperature, the relative risk of DCBs was increased no matter when the temperature went lower or higher from 10 °C. The highest risk of DCBs was found when the temperature reached about 26 °C (Fig. 2F). For the population aged 46–59 years, the relative risk of DCBs was increased with temperature increased (Fig. 2G).
**Table 1  Description of daily emergency department visits for dog and cat bites and meteorological factors during 2016–2020.**

| Variable | Mean ± SD | Min | Max | N (weighted %) |
|---|---|---|---|---|
| Daily number | 14.7 ± 5.8 | 1 | 39 | – |
| Sex | | | | |
|     Male | 7.2 ± 3.4 | 0 | 23 | 13,085 (48.72) |
|     Female | 7.5 ± 3.6 | 0 | 22 | 13,772 (51.28) |
| Age | | | | |
|     ≤14 | 2.4 ± 1.8 | 0 | 10 | 4,419 (16.45) |
|     15–21 | 1.2 ± 1.1 | 0 | 6 | 2,180 (8.12) |
|     22–45 | 5.6 ± 2.9 | 0 | 17 | 10,305 (38.37) |
|     46–59 | 3.1 ± 2.0 | 0 | 12 | 5,633 (20.97) |
|     ≥60 | 2.4 ± 1.7 | 0 | 10 | 4,320 (16.09) |
| $T^a$mean (°C) | 17.4 ± 8.6 | −5.2 | 33.9 | – |
| $DTR^b$ (°C) | 7.4 ± 3.5 | 0.9 | 19.0 | – |
| $AP^c$mean (hPa) | 1,016.1 ± 9.1 | 986.2 | 1,039.7 | – |
| $RH^d$ (%) | 80.2 ± 11.2 | 38.0 | 100.0 | – |
| $WS^e$max (m/s) | 4.6 ± 1.3 | 1.8 | 10.8 | – |
| $WS^e$extreme (m/s) | 8.2 ± 2.4 | 3.2 | 21.4 | – |
| Precipitation (cm) | 4.1 ± 12.3 | 0.0 | 263.5 | – |
| Sunshine (hour) | 4.6 ± 4.2 | 0.0 | 12.8 | – |

**Notes.**
[a]temperature;[b]diurnal temperature range;[c]atmospheric pressure;[d]relative humidity;[E]wind speed.

Population aged over 60 years, the relative risk of DCBs was increased with the temperature rising from 7.6 °C (Fig. 2H).

The 3-D plots show the non-linear relationships between mean temperature and the relative risk of dog and cat bites over a lag of 0–7 days (Fig. 3A).

## DISCUSSION

In this study, we observed a J-shaped relationship between the relative risk of DCBs and ambient temperature and confirmed that the case number of DCBs increased with the increase in temperature. The relationship between meteorological factors and diseases has been well-studied worldwide (*Martinez-Solanas et al., 2018*) and is usually V-, U-, or J-shaped. As temperatures fall below or above certain thresholds, the number of cases or deaths would increase (*Song et al., 2018*). These hot and cold effects are usually quantified once the thresholds are fixed at specific values Yue (*Zhang et al., 2014*). Similar exposure-response relationships were found between ambient temperature and ED visits due to dog bites, basically in the 'U'-shaped version. However, the estimated spline curve of ED visits due to dog bites showed a lower threshold temperature (*Zhang et al., 2017*).

The ED is one of the busiest departments in the hospital, receiving a large number of patients every day, including accidental injuries, emergency attacks, *etc.* In the event of an emergency, the ED is the first department to receive and treat patients and is able to respond and deal with the emergency quickly. The emergency department also provides

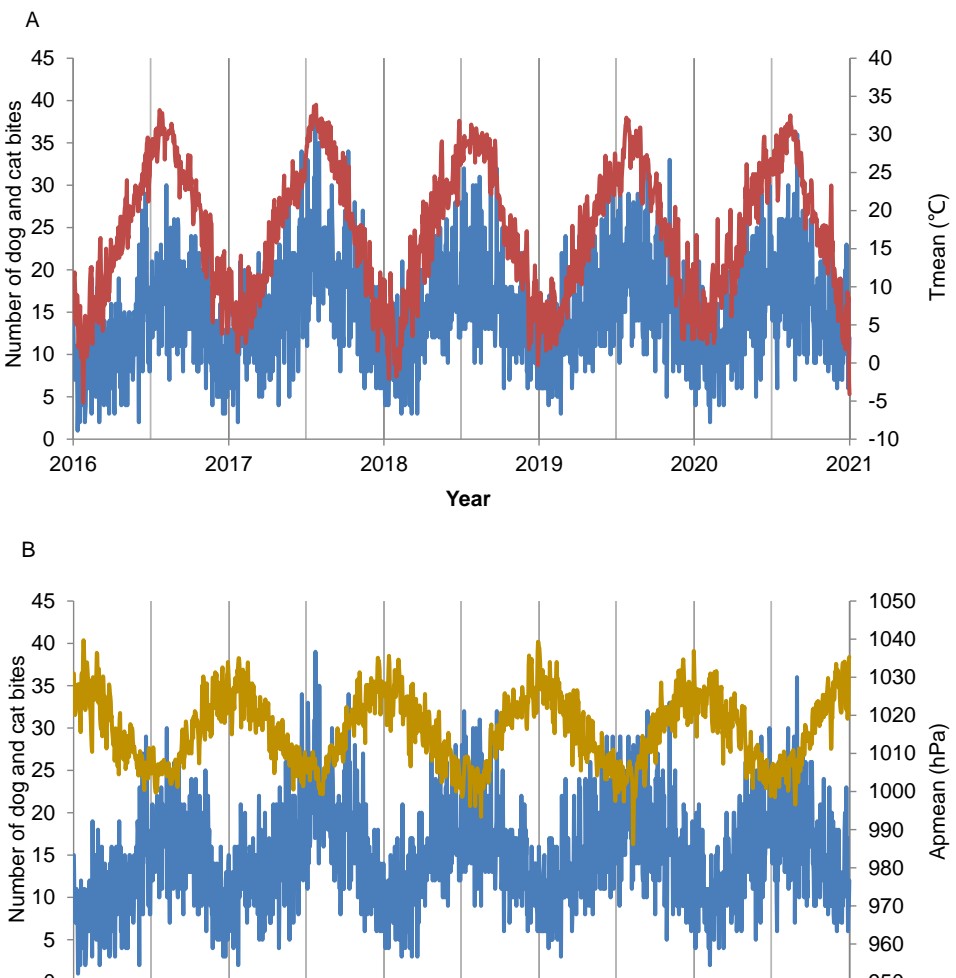

**Figure 1** Correlation of the distribution of daily emergency department visits for dog and cat bites with mean temperature (Tmean) and mean atmospheric pressure (APmean) from years 2016 to 2020.

24-hour medical services and consultations for community residents to ensure the safety and health of residents (*Greenwood-Ericksen & Kocher, 2019*). We observed significant acute effects (within 0–7 days) of ambient temperature on daily ED visits to the DCBs. Especially in hot weather, the incidence rate of DCBs was higher. The underlying causes of this discrepancy were unclear. This is most likely due to, (1) high temperature can directly or indirectly affect the cerebral perfusion pressure and cerebral blood flow of animals leading to their temperament instability, and also can suppress thyroid function, leading to hypothyroidism and mood disorders (*Van Lieshout et al., 2003*). (2) Exposure to high temperatures can lead to significant increases in core body temperature, heart rate, and cerebrovascular resistance (*Nybo, Rasmussen & Sawka, 2014*; *Wilson et al., 2006*). However, in cold weather, people have fewer outdoor activities with walking pets and wear heavy

**Table 2  Spearman's correlation coefficients between daily emergency department visits for dog and cat bites and meteorological factors.**

| Variable | Tmean | DTR | APmean | DAPR | RH | WSmax | WSextreme | Precipitation | Sunshine |
|---|---|---|---|---|---|---|---|---|---|
| Daily number | 0.588** | 0.093** | −0.494** | −0.269** | −0.061** | −0.012 | −0.022 | −0.165** | 0.242** |
| $N_{dog}$ | 0.503** | 0.123** | −0.414** | −0.233** | −0.031 | −0.045 | −0.054* | −0.155** | 0.189** |
| $N_{cat}$ | 0.339** | −0.016 | −0.296** | −0.151** | −0.083** | 0.063** | 0.056* | −0.072** | 0.180** |
| T mean | | −0.094** | −0.894** | −0.385** | 0.112** | 0.112** | 0.101** | 0.010 | 0.228** |
| DTR | | | 0.154** | 0.060** | −0.499** | −0.125** | −0.152** | −0.517** | 0.556** |
| AP mean | | | | 0.249** | −0.251** | −0.170** | −0.151** | −0.168** | −0.103** |
| DAPR | | | | | 0.020 | 0.192** | 0.208** | 0.183** | 0.169** |
| RH | | | | | | −0.082** | −0.096** | 0.626** | −0.589** |
| WS max | | | | | | | 0.931** | 0.140** | 0.083** |
| WS extreme | | | | | | | | 0.182** | 0.056* |
| Precipitation | | | | | | | | | −0.593** |

Notes.

N, case number of emergency department visits for animal bites; T, temperature; DTR, diurnal temperature range; AP, atmospheric pressure; DAPR, diurnal atmospheric pressure range; RH, relative humidity; WS, wind speed.

*$p < 0.05$.

**$p < 0.01$.

clothes, and therefore, the incidence rate is reduced. However, outpatient visits were less affected by these influencing factors. Therefore, meteorological factors are considered to be a good indicator of emergency disease occurrence.

We observed age-associated differences in exposure-response curves, with the risk of being DCBs for people aged 15–21 almost unaffected by temperature, probably because the population aged 15–21 years may spend most of their time in school and spend so little time with domestic pets that their risk of injury was reduced. Population aged 22–45 years were the majority and the relative risk of DCBs was associated with both high and low temperatures. Generally, this age group is economically independent and can raise pets by themselves, and they may need pets for spiritual comfort (*Powell et al., 2018*).

The current study revealed that the temperature-DCBs association varied by gender, such that females were affected by heat more broadly than males. It can be hypothesized that the dimorphism in human gender responses to animal attacks is due to biological differences between males and females. Indeed, Lovick's study found that female-secreted progesterone acts on the panic locus (gray matter around the midbrain aqueduct), leading to increased reactivity to animal aggression (*Lovick, 2014*). A study by *Mishor et al. (2021)* found that a compound known as hexadecanal (HEX) modulates human aggression in a gender-specific manner. HEX works by modulating the functional connectivity of the aggressive brain network, resulting in increased connectivity in males and decreased connectivity in females. At the same time, women wear thin clothes in summer and their skin is more exposed, which increases the risk of being bitten by dogs.

We observed age-associated in exposure-response curve, with subjects aged ≥46 years being more sensitive to hot temperatures, whereas 22–45 years were sensitive to both hot and cold. This is consistent with some relevant studies, which show that older people are more susceptible to high temperatures (*de'Donato et al., 2015*; *Ma et al., 2019*). This may be because the global population is aging and the prevalence of age-associated disease is

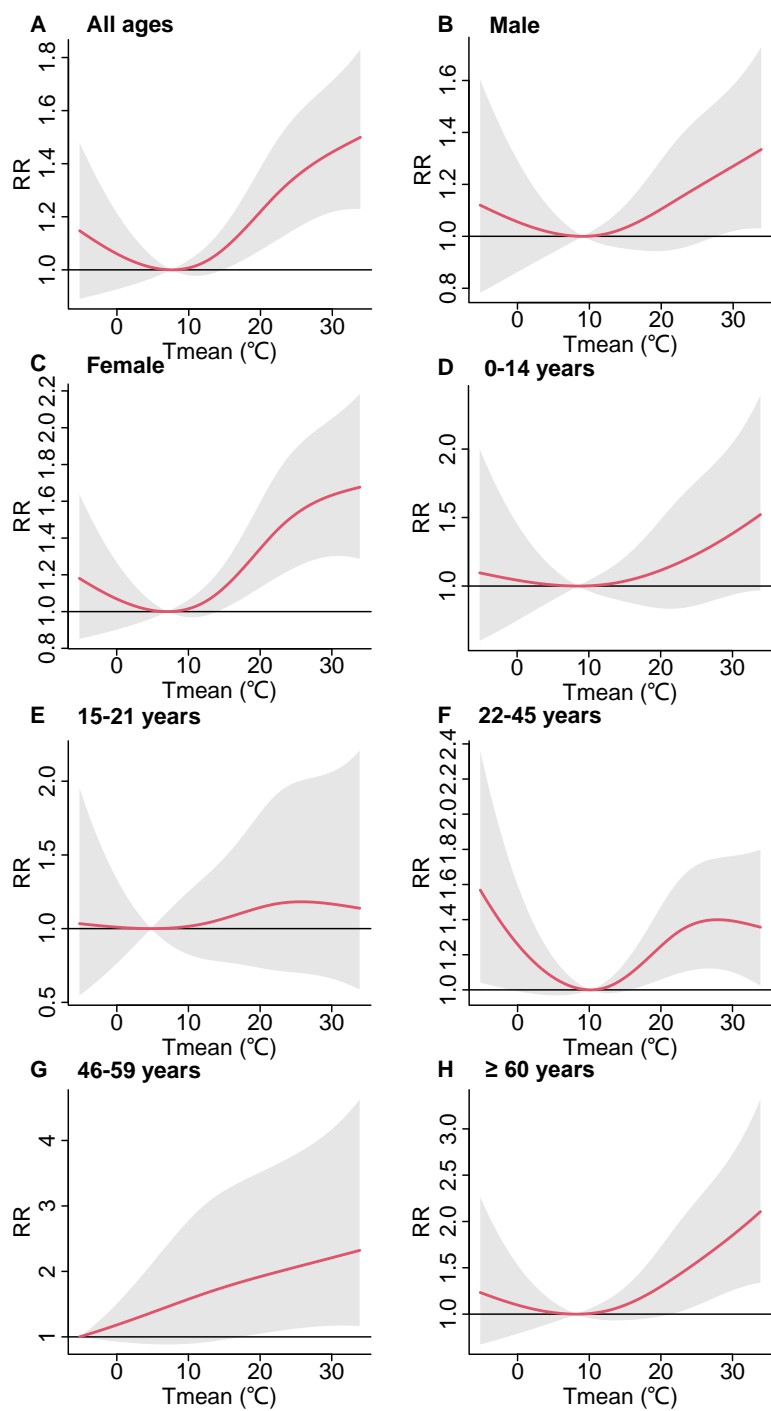

**Figure 2** Cumulative associations between temperature and emergency department visits for dog and cat bites over lag 0–7 days by gender and age groups during 2016–2020.

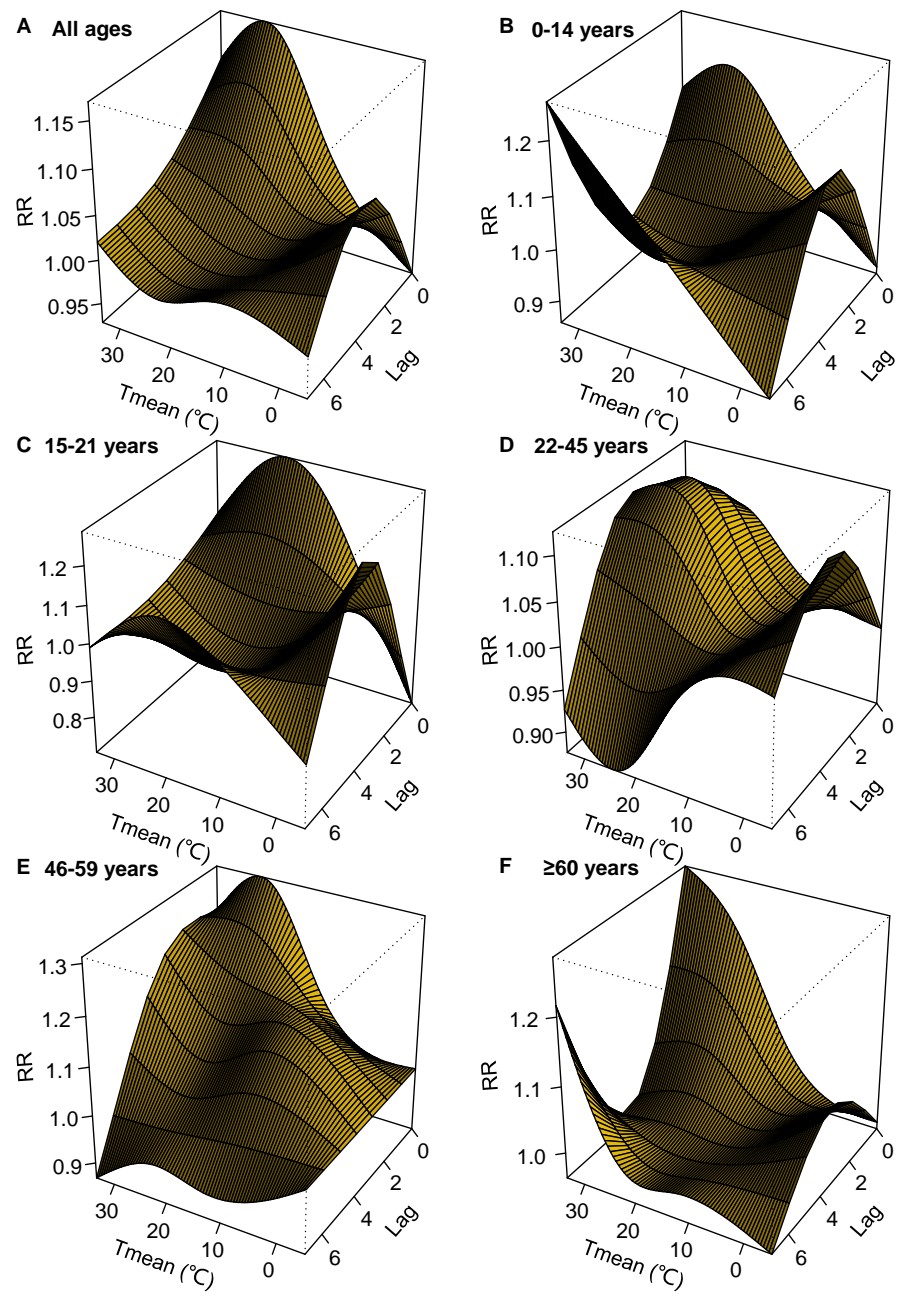

**Figure 3** 3-D plots for the relative risk (RR) of emergency department visits for dog and cat bites associated with mean temperature (Tmean) over lag 0–7 days produced by the distributed lag non-linear mode (DLNM).

on the rise (*Suzman et al., 2015*). At the same time, global climate change poses multiple risks of health hazards for older people, such as the inability to adapt to sudden changes in their surroundings (*Chang et al., 2022*). Individuals aged ≥46 years have a decreased ability to regulate their body temperature and are less able to perceive changes in body temperature during hot weather. As a result, they may not be able to perceive changes in

body temperature or take cooling measures in a timely manner, which also increases the risk of dog bites (*Chambers & Vukmanovic-Stejic, 2020*; *Millyard et al., 2020*). In addition, older persons may have cognitive impairment and mobility problems, which can impair their judgment and reaction ability and make them more vulnerable to attacks by dogs and cats (*Meade et al., 2020*). Therefore, in hot weather, older people can overcome heat stress and improve heat tolerance through appropriate exercise, passive heating, and behavioral adaptation (*Tan, Chin & Low, 2020*). They should also pay special attention to their physical conditions and take timely cooling measures (*Rudnicka et al., 2020*). Of course, family members and caregivers should also enhance protection and care for the elderly to avoid being bitten by dogs and cats.

All organisms are influenced by factors in their surrounding environment. Adverse meteorological factors may have critical effects on the physiological function and behavior of mammals, including humans (*Knapp & Huang, 2022*; *Nakamura & Morrison, 2022*). *In vivo* physiological studies have shown that the hypothalamic-medullary circuitry functions in response to environmental stressors by interacting with other neural circuits as well as physiological systems and that it not only regulates basal body temperature, but also controls a wide range of autonomic and somatomotor responses to heat, psychology, and stress (*Morrison & Nakamura, 2019*; *Nakamura, Nakamura & Kataoka, 2022*; *Tan & Knight, 2018*; *Van Hook, 2020*). Combine this with the American Veterinary Medical Association's assertion that dog and cat bites are primarily a response to the environment or something, such as heat, stressful situations, shock, consternation, or threat (*AVMA, 2023*). It is reasonable to hypothesize that stressors such as meteorological factors act on the hypothalamic-medullary circuits of humans or dogs and cats and that the hypothalamic-medullary circuits trigger cat and dog bites through a series of neurophysiological responses that control autonomic and somatic movements. However, the specific link between meteorological factors and cat and dog bites has yet to be verified through methods such as randomized double-blind exposure experiments in animals or humans.

The effects of ambient temperature on DCBs of ED visits have different delayed structures, which are similar to and different from other diseases in emergency department visits in China (*Hu et al., 2018*; *Zhang et al., 2014*). In addition, a statistically significant association was observed for some, but not all, lag structures of ambient temperature. Therefore, further research is needed to elucidate the lagged structure and extent of this effect.

## Limitations

A few limitations need to be taken into account in the future study. Some likely attributable risk factors such as animal breed, gender, body size, and training as well as victim's behavioral characteristics were not included in the study due to hard quantification and control that may confound the results from collected data and analytic methods. Another potential limitation of this study is data on DCBs were derived from patients who visited a hospital, those small pieces of data were missing from people who did not go to the hospital because they did not think their injuries were serious. Nevertheless, our study was based

on the theoretical basis of the influence of ambient temperature on human and animal mental/behavioral abnormalities.

## CONCLUSIONS

We observed a significant association between exposure to hot ambient temperatures and increased ED visits for DCBs in different gender and age groups in Shanghai, China. The extremely high temperature increased the risk of DCBs, particularly for females and people aged over 46 years. These results indicate that ambient temperature is an important environmental hazard factor in ED visits for DCBs in Shanghai. These findings help establish public health preparedness and interventions to minimize the adverse effects of meteorological factors on DCBs.

## ACKNOWLEDGEMENTS

We thank the anonymous reviewers for their useful comments.

### Funding

This work was supported by the project supported by the Science and Technology Commission of Jinshan District (No. 2018-3-06) and the Jinshan District Health Committee (No. JSKJ-KTGW-2022-05). The funders had no role in study design, data collection and analysis, decision to publish, or preparation of the manuscript.

### Grant Disclosures

The following grant information was disclosed by the authors:
Science and Technology Commission of Jinshan District:  2018-3-06.
Jinshan District Health Committee:  JSKJ-KTGW-2022-05.

### Competing Interests

The authors declare there are no competing interests.

### Author Contributions

- Pei Pang conceived and designed the experiments, performed the experiments, analyzed the data, prepared figures and/or tables, and approved the final draft.
- Xiaoyong Zhou conceived and designed the experiments, analyzed the data, authored or reviewed drafts of the article, and approved the final draft.
- Yabin Hu performed the experiments, prepared figures and/or tables, and approved the final draft.
- Yin Zhang performed the experiments, prepared figures and/or tables, and approved the final draft.
- Baoshi He conceived and designed the experiments, authored or reviewed drafts of the article, and approved the final draft.
- Guoxiong Xu analyzed the data, authored or reviewed drafts of the article, and approved the final draft.

## Human Ethics

The following information was supplied relating to ethical approvals (i.e., approving body and any reference numbers):

The Ethics Committee of Jinshan Hospital of Fudan University approvel to carry out the study within its facilities (Ethical Application Ref: JIEC 2021-S38).

## Data Availability

The raw measurements are available in the Supplementary Files.

## Supplemental Information

Supplemental information for this article can be found online at http://dx.doi.org/10.7717/peerj.16758#supplemental-information.

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
