# Peer review of "Time-series analysis of meteorological factors and emergency department visits due to dog/cat bites in Jinshan area, China"

_PeerJ, doi:10.7717/peerj.16758_

## Round 0.1 · original submission · Major Revisions

Please incorporate all comments of reviewers and please submit the revision with a point-to-point rebuttal letter. If possible, please collect more categories of the data of the sample and incorporate them in the revision.

**Language Note:** The review process has identified that the English language must be improved. PeerJ can provide language editing services - please contact us at copyediting@peerj.com for pricing (be sure to provide your manuscript number and title). Alternatively, you should make your own arrangements to improve the language quality and provide details in your response letter. – PeerJ Staff

Reviewer 1 ·

Basic reporting

The paper on “Time-series analysis of meteorological factors and emergency department visits due to dog/cat bites in Jinshan area, China” is written in good order, well-managed, and easily understandable to the readers. It is according to the scientific standards and will attract researchers who are working on meteorological factors that influence dog/cat bites and have knowledge of this interaction. But before going to publications, there are some modifications required to make it more accurate.
1. Abstract: However, I find some inconsistencies in the abstract section, which the author needs to revise the manuscript carefully. Additionally, there are some minor mistakes which need to be revised carefully and improved.
2. Introduction: I suggest revising the manuscript, removing old references, and replacing it with new ones. I also, suggest elaborating the introduction section and adding two more paragraphs. Also, briefly explain the current study.
3. Kindly revise the introduction section and remove such types of mistakes throughout the manuscript.
4. I suggest revising it and clearly mentioning the objectives and hypothesis. Additionally, elaborate on these hypotheses in the discussion section briefly.
5. Discussion:
Overall, the discussion section could be stronger and needs major revision. Elaborate on this section by mentioning more references and regarding your manuscript.

Experimental design

Materials and methods: Kindly mention the full name here and revise throughout the manuscript. Also, make consistency throughout the manuscript.

Validity of the findings

No comments

Additional comments

• Overall, I suggested a major revision of this manuscript

Reviewer 2 ·

Basic reporting

While professional English was used througout, there is a need to revisit clarity of sentences. For instance in the introduction, line 44 it says "Dog and cat bites has arisen out of serious social and public health problems for human beings around the world" Perhaps the authors meant this as "Dog and cat bites are now serious social and public health problems" I would suggest the authors have a re-check of statements for clarity"

another example is on line 57 "Previous studies have mostly focused on the social and cultural reasons of dog/cat-caused injuries (DCBs), attributing them to human or animal misbehaviors such as encroachment on their territory, breeding, interfering with their eating, and failing to sterilize them." There is a need for clarity here by rewriting the sentence

Otherwise the paper is well structured with readable tables and figues

Experimental design

No comment

Validity of the findings

While the results are interesting and consitent with the growing body of evidence on weather and health condition of humans , the correlations of DCB and temperature require more investigation as their likely cause is behavioural. This is what the study was not able to establish beyond speculation, even if the correlations are significant. Meteorological factors may be important, but how does these affect animal aggressive behaviour? Another complicating factor is the health status of the animal which may affect behaviour. This is noted if the DCB is suspected due to rabies infection.

Some statements require more evidence such as
"This may be explained by the fact that women may be
less defensive and less alert to the environment than men, which may make women easier targets
or dog attacks. At the same time, women wear thin clothes in summer and their skin is more
exposed, which increases the risk of being bitten by dogs.

Can the authors support this with evidence from related studies on human reactions to animal aggression with respect to gender? Otherwise this would be construed as a rather sexist opinion.

Line 204

'Decreased ability to detect changes in body temperature.
As a result, they may not be able to sense changes in their body temperature or take timely steps
cool down, which also increases the risk of being bitten by a dog." More evidence is from the medical and public health literature to link this with dog bites. As global climate change poses a health hazard risk to senior citizens, the authors can cite more similar literature.

Additional comments

While the authors cited the limitations of their study in the lack of data on
"animal breed, gender, body size, and training as well as victim’s behavioral characteristics"

It is these factors that really will make the sudy more convincing in relating meteorological conditions with DCB, which are caused or effected by weather on behavioural responses both by the animal and the human patient. I suggest that they try to get data on this.

---

## Round 0.2 · accepted · Accept

All comments are duly incorporated, and the paper is improved after revision. So, it is accepted for publication.

Reviewer 2 ·

Basic reporting

They have adopted the suggestions of the reviewers and made the necessary changes in English usage.

Experimental design

No comment

Validity of the findings

No comment

Additional comments

The additional suggestions for including more evidence support for their thesis from related literature was made.